# LongLoRA: Efficient Fine-tuning of Long-Context Large Language Models

**Yukang Chen** [1]     **Shengju Qian** [1]     **Haotian Tang** [2]     **Xin Lai** [1]

**Zhijian Liu** [2]     **Song Han** [2,3]     **Jiaya Jia** [1]

[1]CUHK     [2]MIT     [3]NVIDIA

## Abstract

We present LongLoRA, an efficient fine-tuning approach that extends the context sizes of pre-trained large language models (LLMs), with limited computation cost. Typically, training LLMs with long context sizes is computationally expensive, requiring extensive training hours and GPU resources. For example, training on the context length of 8192 needs $16\times$ computational costs in self-attention layers as that of 2048. In this paper, we speed up the context extension of LLMs in two aspects. On the one hand, although *dense global* attention is needed during inference, fine-tuning the model can be effectively and efficiently done by *sparse local* attention. The proposed shifted sparse attention ($S^2$-Attn) effectively enables context extension, leading to non-trivial computation saving with similar performance to fine-tuning with vanilla attention. Particularly, it can be implemented with only *two lines of code* in training, while being optional in inference. On the other hand, we revisit the parameter-efficient fine-tuning regime for context expansion. Notably, we find that LoRA for context extension works well under the premise of trainable embedding and normalization. LongLoRA combines this improved LoRA with $S^2$-Attn. LongLoRA demonstrates strong empirical results on various tasks on Llama2 models from 7B/13B to 70B. LongLoRA extends Llama2 7B from 4k context to 100k, or Llama2 70B to 32k on a single $8\times$ A100 machine. LongLoRA extends models' context while retaining their original architectures, and is compatible with most existing techniques, like Flash-Attention2. In addition, we further conduct supervised fine-tuning with LongLoRA and our long instruction-following LongAlpaca dataset. All our code, models, dataset, and demo are available at github.com/dvlab-research/LongLoRA.

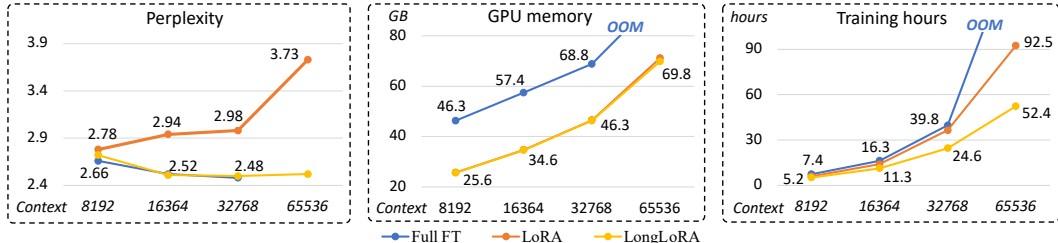

Figure 1: **LongLoRA** closes the accuracy gap that between conventional LoRA and full fine-tuning, while still maintaining up to $1.8\times$ lower memory cost than full fine-tuning. Furthermore, LongLoRA improves the training speed of LoRA by up to $1.8\times$ with $S^2$-Attn. Llama2-7B are fine-tuned to various context lengths with Flash-Attention2 (Dao, 2023) and DeepSpeed (Rasley et al., 2020) stage 2 and evaluated on the proof-pile (Azerbayev et al., 2022) test set in perplexity.

## 1 Introduction

Large language models (LLMs) are typically trained with a pre-defined context size, such as 2048 tokens for LLaMA (Touvron et al., 2023a) and 4096 tokens for Llama2 (Touvron et al., 2023b).

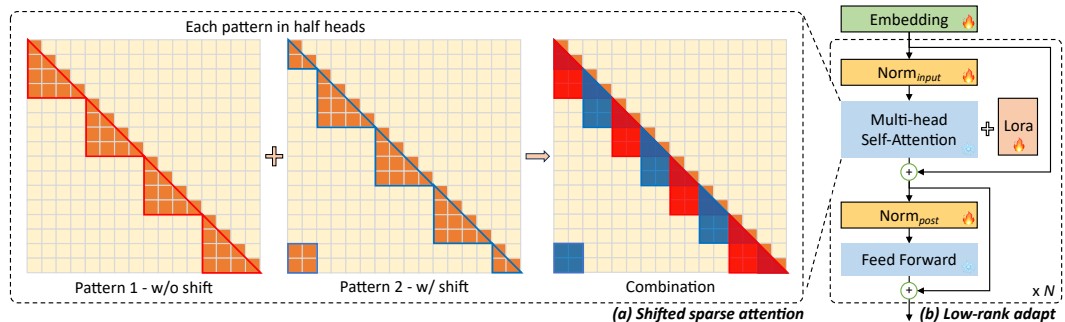

Figure 2: **Overview of LongLoRA**. We introduce Shifted Sparse Attention ($S^2$-Attn) during fine-tuning. The trained model retains original standard self-attention at inference time. In addition to training LoRA weights in linear layers, LongLoRA further makes embedding and normalization layers trainable. This extension is pivotal for context extension, and only introduces a minimal number of additional trainable parameters.

However, the pre-defined size limits LLMs in many applications, like summarizing long documents or answering long questions. To resolve this limitation, some recent works (Chen et al., 2023; Tworkowski et al., 2023; Mohtashami & Jaggi, 2023) train or fine-tune LLMs to longer context. However, training an LLM from scratch with long sequences poses computational challenges, and fine-tuning an existing pre-trained LLM is also considerably expensive. For instance, Position Interpolation (Chen et al., 2023) spent 32 A100 GPUs to extend LLaMA models from 2k to 8k context, and 128 A100 GPUs for longer context fine-tuning. FOT (Tworkowski et al., 2023) used 32 TPUs for standard transformer training and 128 TPUs for LongLLaMA. These computation resources are typically unaffordable for common researchers, which naturally leads us to question: can we extend the context window of LLMs efficiently?

One straightforward approach is to fine-tune a pre-trained LLM via low-rank adaptation (LoRA) (Hu et al., 2022). LoRA modifies the linear projection layers in self-attention blocks by utilizing low-rank matrices, which are generally efficient and reduce the number of trainable parameters. However, our empirical findings indicate that training long context models in this manner is neither sufficiently effective nor efficient. In terms of *effectiveness*, plain low-rank adaptation results in a high perplexity in long context extension, as in Table 2. Increasing the rank to a higher value, *e.g.*, rank = 256, does not alleviate this issue. In terms of *efficiency*, regardless of whether LoRA is employed or not, computational cost increases dramatically as the context size expands, primarily due to the standard self-attention mechanism (Vaswani et al., 2017). As shown in Figure 1, even with LoRA, the training hours for the standard Llama2 model increase substantially when the context window expands.

In this work, we introduce LongLoRA, an efficient fine-tuning approach that extends the context windows of pre-trained LLMs, *e.g.*, Llama2 (Touvron et al., 2023b). LoRA (Hu et al., 2022) uses low-rank weight updates to approximate full fine-tuning. Similarly, we find that short attention is also able to approximate long context during training. We present shifted sparse attention ($S^2$-Attn) as an efficient substitute for standard self-attention. As shown in Figure 2, we split context length into several groups and conduct attention in each group individually. In half attention heads, we shift the tokens by half group size, which ensures the information flow between neighboring groups. For example, we use $S^2$-Attn with group size 2048 to approximate the total 8192 context length training. This shares a high-level spirit with Swin Transformer (Liu et al., 2021).

Models fine-tuned via $S^2$-Attn retain the original attention architecture during inference. This facilitates most existing optimization and infrastructure. Techniques for common LLMs can also be applied to ours. For example, Flash-Attention2 (Dao et al., 2022; Dao, 2023) is compatible with our method in both training and inference time. The reason behind this is that short attention resembles the attention scheme in the pre-training stage of LLMs. Other efficient attentions, *e.g.*, dilated or sparse attention, have a large gap to the standard style and do not work well like ours, as in Table 6.

We empirically show that learnable embedding and normalization layers are the key to unlocking long context LoRA fine-tuning, in Table 2. Embedding and normalization layers take up a small

Figure 3: **Illustration of $S^2$-Attn**. It involves three steps. First, it splits features along the head dimension into two chunks. Second, tokens in one of the chunks are shifted by half of the group size. Third, we split tokens into groups and reshape them into batch dimensions. Attention only computes in each group in ours while the information flows between groups via shifting. Potential information leakage might be introduced by shifting, while this is easy to prevent via a small modification on the attention mask. We ablate this in the variant 2 in Section B.3 in the appendix.

proportion of parameters in the entire LLM. For example, embedding has ($< 2\%$) parameters, and normalization has ($\leq 0.004\%$) parameters in Llama2 7B. This ratio decreases for even larger LLMs.

In experiments, we show that LongLoRA is effective and efficient. We present experimental results of extending the context window for Llama2 7B, 13B, and 70B. Following the experimental settings of Position Interpolation (Chen et al., 2023), we fine-tune models with proper position embeddings. The trained models achieve comparable performance to the full-attention and fully fine-tuned results, while the computational cost is much less as shown in Figure 1. LongLoRA can fine-tune Llama2 7B up to 100k context, or a 70B model up to 32k, on a single $8\times$ A100 machine.

In addition, we present a solution for supervised fine-tuning (SFT) with our self-collected long instruction-following dataset, LongAlpaca. Our LongLoRA models are further fine-tuned with long questions and the corresponding answers. We design various types of questions for technical papers, science fiction, and other books. SFT is important for improving the chat ability of LLMs. We introduce our SFT settings in Section B.6 in the appendix.

## 2 RELATED WORK

**Long-context Transformers.** A large body of research has been developed to increase the context length of transformers. Some of these approaches are retrieval-based (Karpukhin et al., 2020; Izacard et al., 2022; Guu et al., 2020), which augment language models via fetching related documents and including the retrieved results into contexts. Our work is complementary to these works, as our attention mechanism is unmodified during inference. Many works modify multi-head attention to be approximated ones (Wang et al., 2020; Beltagy et al., 2020; Zaheer et al., 2020; Kitaev et al., 2020; Bulatov et al., 2022; Ding et al., 2023; Qiu et al., 2020). They alleviate the quadratic complexity of the self-attention computation. For example, Longformer (Beltagy et al., 2020) and BigBird (Zaheer et al., 2020) use sparse attention to handle long sequences. Other works (Wu et al., 2022; Bulatov et al., 2022) utilize memory mechanisms as a compression on past inputs, to look up relevant tokens. One limitation of these works is that these compressions have a large gap to full attention, making it infeasible to fine-tune pre-trained LLMs. Although our work also involves an approximation of attention mechanism, it has a similar shape and a small gap to standard attention. This enables fine-tuning pre-trained LLMs on $S^2$-Attn and maintain full attention during inference.

**Long-context LLMs.** LLMs are typically pre-trained with a pre-defined context length, such as 2048 for LLaMA (Touvron et al., 2023a) and 4096 for Llama2 (Touvron et al., 2023b). Training LLMs with long context from scratch is prohibitively expensive for most researchers. Recently, several works have tried to extend the context length of LLMs via fine-tuning. Position Interpolation (Chen et al., 2023) modifies rotary position encoding (Su et al., 2021) and extends the context length of LLaMA to 32768. Focused Transformer (Tworkowski et al., 2023) utilizes contrastive learning to train LongLLaMA. Both of them rely on full fine-tuning, which is computationally expensive (128 A100 GPUs / 128 TPUv3 for training). Landmark attention (Mohtashami & Jaggi, 2023) is an

Table 1: **Effectiveness of S$^2$-Attn under different context lengths**. 'Short' means 1/4 of the target context length, while 'Long' equals to the target context length. Models are fully fine-tuned upon a Llama2 (Touvron et al., 2023b) model with 7B parameters on the RedPajama (Computer, 2023) dataset. Results are tested in perplexity on PG19 (Rae et al., 2020) validation split.

| Setting | Position Embedding | Training | | Target Context Length | | |
|---|---|---|---|---|---|---|
| | | Attention | Shift | 8192 | 16384 | 32768 |
| Full Attn | | Long | - | 8.02 | 8.05 | 8.04 |
| Short Attn | PI (Chen et al., 2023) | Short | ✗ | 8.29 | 8.83 | 9.47 |
| S$^2$-Attn | | Short | ✓ | 8.04 | 8.03 | 8.08 |

efficient approach, but somewhat lossy. It compresses long context inputs into retrieved tokens. Our method saves substantial fine-tuning costs, while preserving the quality of the original attention. Ours maintain full access to the entire input via unmodified attention during inference.

Some literature focuses on the position embedding modification of LLMs for long context extension, including Position Interpolation (Chen et al., 2023), NTK-aware (ntk, 2023), Yarn (Peng et al., 2023), positional Skipping (Zhu et al., 2023), and methods based on out-of-distribution analysis (Han et al., 2023). Our method focuses on efficient fine-tuning and retaining the original architecture during inference, which is orthogonal to these position embedding methods.

**Efficient Fine-tuning.** This work is based on LoRA (Hu et al., 2022), a classical efficient fine-tuning approach. In addition to LoRA (Hu et al., 2022), there are many other parameter-efficient fine-tuning methods, including prompt tuning (Lester et al., 2021), prefix tuning (Li & Liang, 2021), hidden state tuning (Liu et al., 2022), bias tuning (Zaken et al., 2022), and masked weight learning (Sung et al., 2021). Input-tuning (An et al., 2022) introduces an adapter to tune input embedding. Although the input embedding layers are also trainable in ours, this is not enough for long context extension. We make a comprehensive analysis on layer types in experiments, in Table 2. Existing work (Chen et al., 2022) shows sparse masks can effectively save training costs and avoid performance drops.

## 3 LONGLORA

### 3.1 BACKGROUND

**Transformer.** LLMs are typically built with transformers. Taking Llama2 (Touvron et al., 2023b) for example, as shown in Figure 2, an LLM model consists of an embedding input layer and a number of decoder layers. Each decoder layer comprises a self-attention module. It maps input features into a set of queries, keys, and values $\{q, k, v\}$, via linear projection layers with weight matrices $\{W_q, W_k, W_v\}$. Given $\{q, k, v\}$, it computes the outputs $o$ as

$$o = \text{softmax}(qk^T)v \tag{1}$$

The outputs are then projected by a linear layer with a weight matrix $W_o$. And MLP layers are followed. Before and after self-attention modules, layer normalization (Ba et al., 2016) is applied. A final normalization is conducted after all decoder layers.

For long sequences, self-attention struggles with computation cost, which is quadratic to the sequence length. This dramatically slows down the training procedure and increases GPU memory costs.

**Low-rank Adaptation.** LoRA (Hu et al., 2022) hypothesizes that the weight updates in pre-trained models have a low intrinsic rank during adaptation. For a pre-trained weight matrix $W \in \mathbb{R}^{d \times k}$, it is updated with a low-rank decomposition $W + \Delta W = W + BA$, where $B \in \mathbb{R}^{d \times r}$ and $A \in \mathbb{R}^{r \times k}$. The rank $r \ll min(d, k)$. During training, $W$ is frozen with no gradient updates, while A and B are trainable. This is the reason why LoRA training is much more efficient than full fine-tuning.

In the Transformer structure, LoRA only adapts the attention weights $(W_q, W_k, W_v, W_o)$ and freezes all other layers, including MLP and normalization layers. This manner is simple and parameter-efficient. However, we empirically show that only low-rank adaptation in attention weights does not work for long context extension.

---

**Algorithm 1:** Pseudocode of S$^2$-Attn in PyTorch-like style.

---

```
# B: batch size; S: sequence length or number of tokens; G: group size;
# H: number of attention heads; D: dimension of each attention head

# qkv in shape (B, N, 3, H, D), projected queries, keys, and values
# Key line 1: split qkv on H into 2 chunks, and shift G/2 on N
qkv = cat((qkv.chunk(2, 3)[0], qkv.chunk(2, 3)[1].roll(-G/2, 1)), 3).view(B*N/G,G,3,H,D)

# standard self-attention function
out = self_attn(qkv)

# out in shape (B, N, H, D)
# Key line 2: split out on H into 2 chunks, and then roll back G/2 on N
out = cat((out.chunk(2, 2)[0], out.chunk(2, 2)[1].roll(G/2, 1)), 2)
```

---

`cat`: concatenation; `chunk`: split into the specified number of chunks; `roll`: roll the tensor along the given dimension.

---

## 3.2 SHIFTED SPARSE ATTENTION

Standard self-attention costs $O(n^2)$ computations, making LLMs on long sequences high memory cost and slow. To avoid this issue during training, we propose Shifted Sparse Attention (S$^2$-Attn), as shown in Figure 2. In the following, we make a pilot study and explain our design step by step.

**Pilot Study.** In Table 1, we build up a standard baseline that is trained and tested with full attention and fine-tuning, which presents consistently good quality in various context lengths. The first trial is to train with short attention, *only pattern 1* in Figure 2. As we know for a long context, the high cost mainly comes from self-attention modules. Thus, in this trial, since the input is long, we split into several groups in self-attention. For example, the model takes 8192 tokens as input in both the training and testing stages, but self-attention is conducted in each group with a 2048 size. The group number is 4, as ablated in Section B.2 in the appendix. This pattern is efficient but still does not work in a very long context, as shown in Table 1. The perplexity becomes larger as the context length increases. The reason behind this is that there is no information exchange between different groups.

To introduce communication between groups, we include a shifted pattern, as shown in Figure 2. We shift the group partition by half group size in half attention heads. Taking the overall 8192 context length for example, in pattern 1, the first group conducts self-attention from 1$^{st}$ to 2048$^{th}$ tokens. In Pattern 2, the group partition is shifted by 1024. The first attention group begins from 1025$^{th}$ and ends at 3072$^{th}$ tokens, while the first and the last 1024 tokens belong to the same group. We use patterns 1 and 2 in each half self-attention heads respectively. This manner does not increase additional computation costs but enables the information flow between different groups. We show that it gets close to the standard attention baseline in Table 1.

**Consistency to Full Attention.** Existing efficient attention designs can also improve the efficiency of long-context LLMs. However, most of them are not suitable for long-context fine-tuning. Because, these transformers (Qiu et al., 2020; Child et al., 2019), designed for training from scratch, have gaps to the standard full attention, which is used in pre-training. In Table 6, we show that S$^2$-Attn not only enables efficient fine-tuning but also supports full attention testing. Although other attentions can also be used in long context fine-tuning, models must be tested with the attention used during fine-tuning. Shifting prevents models from being over-fitted to specific attention patterns.

**Easy Implementation.** S$^2$-Attn is easy to implement. It involves only two steps: (1) shifting tokens in half attention heads, and (2) transposing features from token dimension to batch dimension. Two lines of code are enough. We provide a PyTorch-style code in Algorithm 1.

## 3.3 IMPROVED LoRA FOR LONG CONTEXT

LoRA (Hu et al., 2022) is an efficient and popular manner for adapting LLMs to other datasets. It saves much trainable parameters and memory cost, compared to full fine-tuning. However, adapting LLMs from short context length to long is not easy. We empirically observe an obvious gap between LoRA and full fine-tuning. As shown in Table 2, the gap between LoRA and full fine-tuning grows as the target context length becomes larger. And LoRA with larger ranks cannot reduce the gap.

Table 2: **Finetuning normalization and embedding layers is crucial for low-rank long-context adaptation**. Llama2 7B (Touvron et al., 2023b) models with the proposed S$^2$-Attn are trained on the RedPajama (Computer, 2023) dataset. The target context length is 32768. '+ Normal / Embed' means normalization or embedding layers are trainable. Perplexity results are evaluated on PG19 (Rae et al., 2020) validation set. For long context adaptation, there is a large performance gap between standard LoRA (Hu et al., 2022) and full fine-tuning. Without trainable normalization or embeddings, larger ranks in LoRA can not close this gap.

| Method | Full FT | LoRA (rank) | | | | | | LoRA (rank = 8) | | |
|---|---|---|---|---|---|---|---|---|---|---|
| | | 8 | 16 | 32 | 64 | 128 | 256 | + Norm | + Embed | + Norm & Embed |
| PPL | 8.08 | 11.44 | 11.82 | 11.92 | 11.96 | 11.97 | 11.98 | 10.49 | 8.29 | 8.12 |

Table 3: Perplexity evaluation on proof-pile (Rae et al., 2020) test split. S$^2$-Attn: Shifted Sparse Attention. LoRA$^+$: improved LoRA. We fine-tune Llama2 (Touvron et al., 2023b) in 7B and 13B model sizes on the RedPajama (Computer, 2023) dataset under 8k-32k context lengths. We show that our method achieves comparable performance to the full attention or full FT baselines, with better efficiency. We use the same training setting as the model evaluated on PG19 (Rae et al., 2020) introduced in Section B.1 in the appendix.

| Size | Training Context Length | LongLoRA S$^2$-Attn | LongLoRA LoRA$^+$ | Evaluation Context Length 2048 | 4096 | 8192 | 16384 | 32768 |
|---|---|---|---|---|---|---|---|---|
| 7B | 8192 | | | 3.14 | 2.85 | 2.66 | - | - |
| | | ✓ | | 3.15 | 2.86 | 2.68 | - | - |
| | | ✓ | ✓ | 3.20 | 2.91 | 2.72 | - | - |
| | 16384 | ✓ | | 3.17 | 2.87 | 2.68 | 2.55 | - |
| | | ✓ | ✓ | 3.17 | 2.87 | 2.66 | 2.51 | - |
| | 32768 | ✓ | | 3.20 | 2.90 | 2.69 | 2.54 | 2.49 |
| | | ✓ | ✓ | 3.35 | 3.01 | 2.78 | 2.61 | 2.50 |
| 13B | 8192 | | | 2.96 | 2.69 | 2.53 | - | - |
| | | ✓ | | 3.01 | 2.74 | 2.57 | - | - |
| | | ✓ | ✓ | 3.04 | 2.77 | 2.60 | - | - |
| | 16384 | ✓ | | 2.99 | 2.72 | 2.53 | 2.40 | - |
| | | ✓ | ✓ | 3.03 | 2.74 | 2.55 | 2.41 | - |
| | 32768 | ✓ | | 3.04 | 2.75 | 2.56 | 2.42 | 2.33 |
| | | ✓ | ✓ | 3.05 | 2.76 | 2.57 | 2.42 | 2.32 |

To bridge this gap, we open embedding and normalization layers for training. As shown in Table 2, they occupy limited parameters but make effects for long context adaptation. Especially for normalization layers, the parameters are only $0.004\%$ in the whole Llama2 7B. We denote this improved version of LoRA as LoRA$^+$ in experiments.

# 4 EXPERIMENT

## 4.1 EXPERIMENTAL SETTINGS

**Models** We extend the pre-trained 7B, 13B, and 70B Llama2 (Touvron et al., 2023b) models. The maximum extended context window sizes are up to 100k for 7B models, 65536 for 13B models, and 32768 for 70B models. The position indices for these models are re-scaled with Position Interpolation (Chen et al., 2023).

**Training Procedure** We follow most training hyper-parameters in Position Interpolation (Chen et al., 2023), except that our batch size is smaller as we use a single $8\times$ A100 GPUs machine in some cases. All models are fine-tuned via the next token prediction objective. We use AdamW (Loshchilov & Hutter, 2019) with $\beta_1 = 0.9$ and $\beta_2 = 0.95$. The learning rate is set to $2 \times 10^{-5}$ for 7B and 13B models, and $10^{-5}$ for 70B models. We also use a linear learning rate warmup. The weight decay is

Table 4: Maximum context length that we can fine-tune for various model sizes on a single 8×A100 machine. We use the same training and evaluation settings as in Table 3. We use Flash-Attention2 (Dao, 2023) and DeepSpeed (Rasley et al., 2020) in stage 3 during fine-tuning. With LongLoRA, the maximum context length for 7B, 13B, and 70B models are 100k, 64k, and 32k respectively. Evaluation on PG19 (Rae et al., 2020) is in Section B.1 in the appendix.

| Size | Training Context Length | Evaluation Context Length | | | | | | |
|------|------|------|------|------|------|------|------|------|
| | | 2048 | 4096 | 8192 | 16384 | 32768 | 65536 | 100,000 |
| 7B | 100,000 | 3.36 | 3.01 | 2.78 | 2.60 | 2.58 | 2.57 | 2.52 |
| 13B | 65536 | 3.20 | 2.88 | 2.66 | 2.50 | 2.39 | 2.38 | - |
| 70B | 32768 | 2.84 | 2.57 | 2.39 | 2.26 | 2.17 | - | - |

Table 5: Topic retrieval evaluation with LongChat (Li et al., 2023). We compare our model to other open-source long-context LLMs. This task involves retrieving target topics from a very long conversation with around 3k, 6k, 10k, 13k, and 16k context lengths. As some questions in the evaluation set are longer than 16k, our model is fine-tuned upon Llama2 13B. It achieves comparable performance to the state-of-the-art LongChat-13B (Li et al., 2023) with a lower fine-tuning cost.

| Evaluation Context | 3k | 6k | 10k | 13k | 16k |
|------|------|------|------|------|------|
| ChatGLM2-6B (Du et al., 2022) | 0.88 | 0.46 | 0.02 | 0.02 | 0.02 |
| MPT-30B-chat (Team, 2023a) | 0.96 | **1.0** | 0.76 | - | - |
| MPT-7B-storywriter (Team, 2023b) | 0.46 | 0.46 | 0.28 | 0.34 | 0.36 |
| LongChat-13B (Li et al., 2023) | **1.0** | **1.0** | **1.0** | **0.98** | 0.9 |
| Ours-13B | **1.0** | 0.98 | 0.98 | **0.98** | **0.94** |

zero. We set the per-device batch size as 1 and gradient accumulation steps as 8, which means that the global batch size equals 64, using 8 GPUs. We train our models for 1000 steps.

**Datasets** We use the Redpajama (Computer, 2023) dataset for training. We evaluate the long-sequence language modeling performance of our fine-tuned models on the book corpus dataset PG19 (Rae et al., 2020) and the cleaned Arxiv Math proof-pile dataset (Azerbayev et al., 2022). We use the test split of PG19 (Rae et al., 2020), consisting of 100 documents. For the proof-pile dataset, we also use the test split of it for evaluation. We follow Position Interpolation (Chen et al., 2023) for proof-pile data processing. We evaluate perplexity by using a sliding window approach with $S = 256$, following (Press et al., 2022).

## 4.2 MAIN RESULTS

**Long-sequence Language Modeling.** In Table 3, we report the perplexity for our models and baseline on proof-pile (Azerbayev et al., 2022) and PG19 datasets. Under certain training context lengths, our models achieve better perplexity with longer context sizes. This indicates the effectiveness of our efficient fine-tuning method. In Table 3, for the same training and evaluation context length cases, the perplexity decreases as the context size increases. By increasing the context window size from 8192 to 32768, for the Llama2 7B model, we observe that the perplexity gets better from 2.72 to 2.50 by -0.22. For Llama2 13B model, we observe that the perplexity reduces by -0.28.

In Table 4, we further examine the maximum context length that we can fine-tune on a single 8×A100 machine. We extend Llama2 7B, 13B, and 70B to 100k, 65536, and 32768 context length respectively. LongLoRA achieves promising results on these extremely large settings. In addition, we find some perplexity degradation on small context sizes for the extended models. This is a known limitation of Position Interpolation (Chen et al., 2023).

**Retrieval-based Evaluation.** We conduct experiments on retrieval in long contexts. In Table 5, we compare our model with other open LLMs on the topic retrieval task introduced in LongChat (Li et al., 2023). This task is to retrieve the target topic from a very long conversation, with lengths varying from 3k, 6k, 10k, 13k, to 16k. As some questions in LongChat (Li et al., 2023) are longer than 16k, we fine-tuned Llama2 13B with a context length of 18k. The training cost is similar to that for 16k.

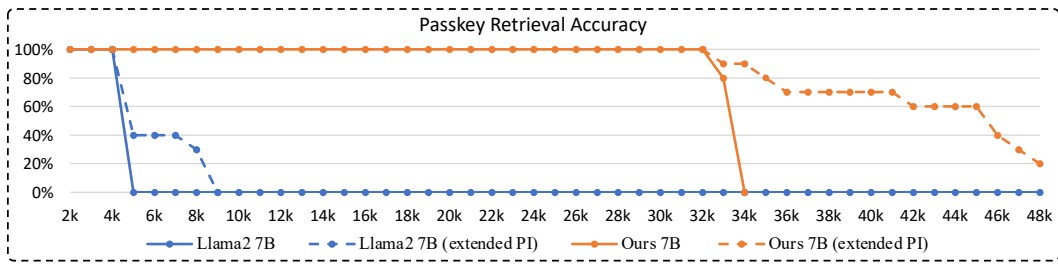

Figure 4: Accuracy comparison on passkey retrieval between Llama2 7B and our 7B model fine-tuned on 32768 context length. Our model presents no retrieval accuracy degradation until 33k or 34k, which exceeds the context length. It can further enhance its capability of long sequence modeling through a straightforward extension of position embeddings, without additional fine-tuning.

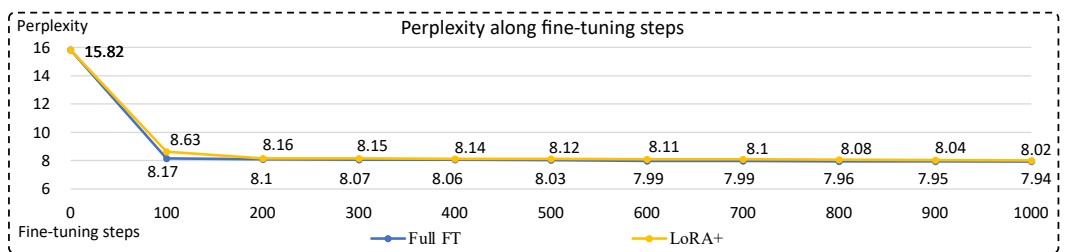

Figure 5: Ablation on fine-tuning steps in both full fine-tuning and LoRA$^+$. We fine-tune Llama2 (Touvron et al., 2023b) 7B with the proposed $S^2$-Attn. The target context length is 8192. We use RedPajama (Computer, 2023) for training and PG19 (Rae et al., 2020) validation set for perplexity testing. Full fine-tuning converges faster than LoRA$^+$ at the beginning, but the final performance gap is small.

Our model achieves comparable performance to LongChat-13B (Li et al., 2023), the state-of-the-art model in this task. Unlike LongChat-13B (Li et al., 2023), which is fully fine-tuned on self-collected long context conversation text, our model is efficiently adapted on RedPajama (Computer, 2023) via next-token generation. Our model even slightly outperforms LongChat-13B in the 16k evaluation.

In Figure 4, we present the passkey retrieval accuracy of our model, following Landmark Attention (Mohtashami & Jaggi, 2023). This task has also been adopted by other literature (Chen et al., 2023; Tworkowski et al., 2023). In this task, the models need to find a random passkey hidden in a long document. We show the document format is in Section A.2 in the appendix. We study Llama2 7B (Touvron et al., 2023b) and our LongLoRA model which fine-tunes Llama2 7B with 32768 context length. We test the passkey retrieval accuracy from 1k to 34k, with an interval of roughly 1k (as the sentence length can not be precisely controlled). For each document length, we test the model 10 times with different random passkey values. Our model achieves reasonable passkey retrieval accuracy until 33k or 34k. Without further fine-tuning, We modify the max position embeddings to 48k in the position interpolation, which is the Ours 7B (extended PI) in Figure 4. We show that this model can handle longer documents by simply extending the position interpolation. As the dashed orange line in Figure 4, the model, fine-tuned on 32k context length, presents moderate retrieval ability (60%-90% accuracy) in the range of 33k to 45k. Even with the position interpolation extended, Llama2 7B suffers from a sharp accuracy degradation (dashed blue line) after the 4k context length.

### 4.3 ABLATION STUDY

In this section, we introduce ablation studies on the number of fine-tuning steps and attention patterns. Other experimental results including ablations on group sizes, attention variants, and efficiency analysis are Section B in the appendix.

**Ablation on Fine-tuning Steps.** We report the relationship between perplexity and fine-tuning steps for a Llama2 7B model extending to the 8192 context length on the PG19 validation set, in

Table 6: **Comparisons among S$^2$-Attn and alternative attention patterns during fine-tuning**. We adapt a Llama2 7B model to 32768 context length with different attention patterns and improved LoRA at training time. We include four typical efficient attention designs, *e.g.*, shift, dilate (Ding et al., 2023), block sparse (Qiu et al., 2020), stride sparse (Child et al., 2019) for comparison. '*cro. heads / layers*' means to swap different attention settings across attention *heads* or sequential *layers*. Taking S$^2$-Attn as an example, '*cro. layers*' is to swap between w/ and w/o shift in sequential self-attention layers. '*only P1/P2*' means all attention heads use pattern 1 (all no shift) or Pattern 2 (all shift) in Figure 2. We visualize the patterns of different attention in Figure 7 in the appendix. For each attention pattern, we evaluate its performance under two protocols. In the first row, we use sparse attention in both training and testing. In the second row, we use full attention for testing.

| Test w/ | S$^2$-Attn | | | | Dilate | Block sparse | Stride sparse |
| Full-Attn | **cro. heads** | *cro. layers* | *only P1.* | *only P2.* | *cro. heads* | *cro. heads* | *cro. heads* |
|---|---|---|---|---|---|---|---|
| ✗ | 8.64 | 8.63 | 9.17 | 9.64 | 8.75 | 11.49 | 32.81 |
| ✓ | 8.12 | 9.70 | 8.39 | 9.81 | 11.78 | 8.30 | 24.03 |

Figure 5. We see that without fine-tuning, at step 0, the model has a limited long context capability, *e.g.*, 15.82 perplexity. We show that the perplexity drops quickly. Full fine-tuning converges faster than low-rank training. They come closer after 200 steps, without a large gap at the end.

**Attention Patterns.** In Table 6, we show the effects of different attention patterns during fine-tuning. We fine-tune a Llama2 7B (Touvron et al., 2023b) model to 32768 context length on Redpajama (Computer, 2023) datasets and evaluate the perplexity on PG19 (Rae et al., 2020) validation set. We first examine the manner of swapping among various settings. For the shift operation we used in LongLoRA, there are three choices: disabling it, shifting between sequential layers, and shifting among attention heads. We show that shifting between layers is acceptable but not the best. In addition, setting all attention heads as pattern 1 or pattern 2 does not work. In addition, we empirically find that shifting left or right has little difference in performance.

We then test other types of efficient attention designs, including dilated attention (Ding et al., 2023), block sparse attention (Qiu et al., 2020), and stride sparse attention (Child et al., 2019). For dilated attention (Ding et al., 2023), we vary the dilate rate from 1 to 2 evenly among attention heads. For block sparse attention (Qiu et al., 2020), we use $n = 4$ block-wise masking matrices in attention heads and move the block left to make it causal. Stride sparse attention (Child et al., 2019) contains both local and stride patterns. These settings share similar computational costs. We visualize these patterns in Figure 7 in the appendix. These attention patterns are invented in training-from-scratch transformers. This experiment is to examine their capability of fine-tuning on pre-trained LLMs (Touvron et al., 2023b), toward long context adaptation. Dilated attention performs well in full fine-tuning but is not well with low-rank adaptation. Fine-tuning with stride sparse attention is harmful. They have a large gap to full attention, which is applied in the pre-training stage.

## 5 CONCLUSION

In this work, we propose LongLoRA that can efficiently extend the context length of LLMs to be significantly larger. LongLoRA has less GPU memory cost and training time than standard full fine-tuning, with minimal accuracy compromise. At the architecture level, we propose $S^2$-Attn to approximate the standard self-attention pattern during training. $S^2$-Attn is easy to implement, requiring only two lines of code. Moreover, models trained via $S^2$-Attn retain the original standard attention architecture during inference, making most pre-existing infrastructure and optimization reusable. At the training level, we bridge the gap between LoRA and full fine-tuning with trainable normalization and embedding. Our method can extend Llama2 7B to 100k context length and 70B model to 32k context length, on a single $8\times$ A100 machine. We also present a long instruction-following dataset, LongAlpaca and conducted supervised fine-tuning with LongLoRA. We believe that LongLoRA is a general method that could be compatible with more types of LLMs and position encodings. We plan to investigate these in future work.

**Acknowledgement** We would like to thank Xiuyu Li and Bohao Peng for the helpful discussions.

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

APPENDIX

## A SETTINGS

### A.1 ENVIRONMENTS

All our experiments are conducted on an $8\times$ A100 machine. We train all models using Py-Torch (Paszke et al., 2019) with the DeepSpeed (Rasley et al., 2020) and Flash-Attention2 (Dao, 2023). By default, we use DeepSpeed (Rasley et al., 2020) in stage 2 and use stage 3 for the maximum context length experiments. Gradient checkpoint is used by default, which is a common technique in the Peft codebase (Mangrulkar et al., 2022). Note that sometimes, $8\times$ A100 GPUs might not be necessary and 3090 Ti GPUs are acceptable, like fine-tuning 7B models to 8192 context size.

### A.2 FORMAT OF PASSKEY RETRIEVAL

We follow existing literature (Mohtashami & Jaggi, 2023; Tworkowski et al., 2023; Chen et al., 2023) for the document format of passkey retrieval. The document has the following format:

```
There is an important info hidden inside a lot of irrelevant text.
Find it and memorize them.  I will quiz you about the important
information there.
The grass is green.  The sky is blue.  The sun is yellow.  Here we
go.  There and back again.  (repeat M times)
The pass key is 12362.  Remember it.  12362 is the pass key.
The grass is green.  The sky is blue.  The sun is yellow.  Here we
go.  There and back again.  (repeat N times)
What is the pass key?  The pass key is
```

The document length varies with the value of `M` and `N`. **12362** is the passkey number to retrieve. It is randomly sampled and varies at each testing time.

## B EXPERIMENTS

### B.1 EVALUATION PERPLEXITY ON PG19 TEST SPLIT.

In Table 14 and Table 15, we present the evaluation results on the PG19 test split. We use the same settings as the models on proof-pile (Azerbayev et al., 2022) evaluation in the paper. Similarly, for a model trained on a certain context length, as the evaluation context length increases, our models achieve better perplexity. Note that the perplexity in Table 14 and Table 15 is higher than that in the proof-pile dataset, as PG19 (Rae et al., 2020) has very different writing styles.

### B.2 ABLATION ON GROUP SIZES.

In Table 7, we provide an ablation study on the group size of the $S^2$-Attn. We experimented with fine-tuning Llama2 7B to 8192 and 16384 context lengths via LongLoRA. The group size varies from $\{1/2, 1/4, 1/6, 1/8\}$ of the target context length. For example, the group size is 1024 for 1/8 of the context length 8192. We find that the 1/2 and 1/4 settings have minor gaps to full attention fine-tuning. Group sizes less than 1/4 would be not good enough. We set the group size as 1/4 of the context length in experiments by default.

Table 7: Ablation on group size. We fine-tune a Llama2 7B model to 8192 and 16384 context lengths via LongLoRA and evaluate on PG19 validation set. We vary the group size of $S^2$-Attn from $\{1/2, 1/4, 1/6, 1/8\}$ of the target context length. 'Full' means the standard full attention.

| Context Length | Full | 1/2 | 1/4 | 1/6 | 1/8 |
|---|---|---|---|---|---|
| 8192 | 8.02 | 8.04 | 8.04 | 8.10 | 8.16 |
| 16384 | 7.82 | 7.84 | 7.86 | 7.94 | 7.98 |

### B.3 ABLATION ON THE VARIANTS OF S$^2$-ATTN.

In Table 8, we ablate some variants of S$^2$-Attn, which are illustrated in Figure 6. Variant 1 is to change the shifting direction from down to up. It shows that the shifting direction has no effect on the perplexity. One concern about S$^2$-Attn is that it moves the last tokens to the front into one group, which might be inconsistent with causal masks. Variant 2 uses individual groups for the shifted tokens, which ablates this concern. Variant 3 swaps the shifted and the original front tokens, which can also ablate the concern. We show that these variants present similar perplexity to ours. We suppose that although there are communications among the front and last tokens, they are originally far away from others while it is limited in the local group. Moreover, S$^2$-Attn is only used for fine-tuning, while we use standard causal masks and full attention during inference. Variant 2 and 3 also work well but involve additional steps to ours.

Table 8: Ablation on the variants of S$^2$-Attn. These variants are illustrated in Figure 6. Similar to the setting in Table 7, we fine-tune a Llama2 7B to 8192 context and evaluate on PG19 validation set.

| Attn | Full | Ours | Variant 1 | Variant 2 | Variant 3 |
|------|------|------|-----------|-----------|-----------|
| PPL  | 8.02 | 8.04 | 8.04 | 8.03 | 8.05 |

Table 9: Evaluation on LongBench (Bai et al., 2023) benchmark. In each column, we highlight the highest value to be bold and the second highest value with underline.

| Model | Avg | Single-Doc QA | Multi-Doc QA | Summarization | Few-shot Learning | Code | Synthetic |
|-------|-----|---------------|--------------|---------------|-------------------|------|-----------|
| GPT-3.5-Turbo | **44.0** | **39.8** | **38.7** | 26.5 | **67.1** | 54.1 | **37.8** |
| Llama2-7B-chat | 31.0 | 24.9 | 22.6 | 24.7 | 60.0 | 48.1 | 5.9 |
| LongChat-v1.5-7B | 34.3 | 28.7 | 20.6 | 26.7 | 60.0 | 54.1 | 15.8 |
| Vicuna-v1.5-7B | 31.9 | 28.0 | 18.6 | 26.0 | 66.2 | 47.3 | 5.5 |
| Ours-7B | 36.8 | 28.7 | 28.1 | **27.8** | 63.7 | **56.0** | 16.7 |

Table 10: Evaluation on LEval (An et al., 2023) open-ended benchmark. We compare various models to GPT-3.5-Turbo and judge win rates via GPT-4.

| Model | Win-rate | Wins | Ties |
|-------|----------|------|------|
| LongChat-7B (Li et al., 2023) | 33.68 | 36 | 56 |
| LongChat-v1.5-7B (Li et al., 2023) | 33.59 | 38 | 53 |
| Vicuna-v1.5-7B (Chiang et al., 2023) | 25.52 | 22 | 54 |
| Ours-7B | **39.06** | **45** | **60** |

### B.4 EVALUATION ON LONG-CONTEXT BENCHMARKS.

We evaluate our method on long-context benchmarks, LongBench (Bai et al., 2023) in Table 9 and LEval (An et al., 2023) in Table 10. We fine-tune Llama2 7B to 16384 context length, with the supervised fine-tuning method and data introduced in Section B.6. We compare our model with GPT-3.5-Turbo and other Llama2-based long-context models, like Vicuna (Chiang et al., 2023) and LongChat (Li et al., 2023) models. It shows that our 7B model presents comparable or even better performance than these Llama2-based long-context models, while ours only takes about 4 hours, about 0.3 billion tokens, on a single 8$\times$ A100 machine.

### B.5 EFFICIENCY ANALYSIS.

In Table 11, we break down the FLOPs of Llama2 7B (Touvron et al., 2023b) into various types of layers, including FFN - feed-forward layers, Proj - projection for queries, values, keys, and attention outputs, Attn - self-attention computation, Others - other layers like embedding, normalization, LLM head. For full attention, the proportion of Attn sharply increases as the context length increases. For

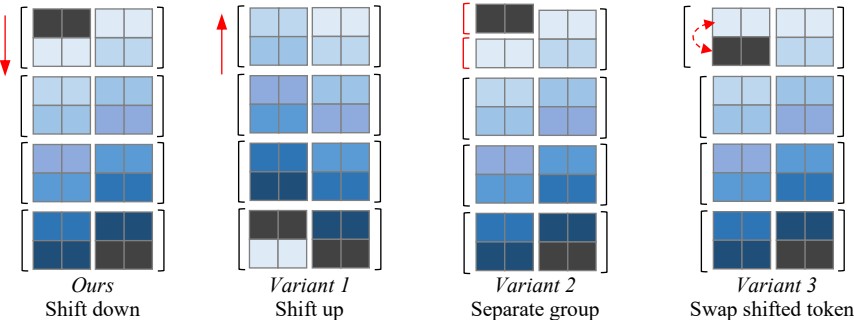

|  |  |  |  |
|---|---|---|---|
| *Ours*
Shift down | *Variant 1*
Shift up | *Variant 2*
Separate group | *Variant 3*
Swap shifted tokens |

Figure 6: Illustration on the variants of our $S^2$-Attn. Variant 1 changes the shifting direction. Variant 2 splits the shifted tokens into one individual group. Variant 3 swaps the shifted tokens with the original front one.

Table 11: FLOPs profiling on various context lengths. We break down the Llama2 7B model into FFN (feed-forward layers), Proj (projection layers for queries, keys, values, and attention outputs), Attn (self-attention kernel), and Others (*e.g.*, embedding, normalization, LLM head). The ratio of attention in the overall model increases as the context length increases. $S^2$-Attn reduces the FLOPs by a large margin, especially when the context length is large.

| Context
Length | $S^2$-Attn | FLOPs (T) | | | | |
|---|---|---|---|---|---|---|
| | | Attn | Proj | FFN | Others | Total |
| 8192 | ✗ | 35.2 | 35.2 | 70.9 | 2.2 | 143.5 |
| | ✓ | 8.8 | | | | 117.1 |
| 16384 | ✗ | 140.7 | 70.4 | 141.8 | 4.3 | 357.2 |
| | ✓ | 35.2 | | | | 251.7 |
| 32768 | ✗ | 562.9 | 140.7 | 283.7 | 8.7 | 996.0 |
| | ✓ | 140.7 | | | | 573.8 |
| 65536 | ✗ | 2251.8 | 281.5 | 567.4 | 17.3 | 3118.0 |
| | ✓ | 562.9 | | | | 1429.1 |

example, Attn has 24.5% of the total FLOPs at the 8192 context length while it increases to 72.2% at the 65536 context length. It decreases to 39.4% when $S^2$-Attn is used.

For the measurement of FLOPs in Table 11, We profiled the context stage FLOPs of Llama2-7B using a batch size of 1 and various context lengths using a third-party tool, torchprofile [1]. The tool traces the computation graph and sums up the FLOPs of each node in the graph (e.g. Q/K/V/O projections, multi-head self-attention, fully-connected layers, and normalization layers).

In Table 12, we compare the training cost among full fine-tuning, plain LoRA (Hu et al., 2022), and LongLoRA. It records details for Figure 1 in the paper. The major difference between LoRA (Hu et al., 2022) and LongLoRA is the $S^2$-Attn. Although there are many FLOPs saving, the peak memory cost has limited difference, because of the highly optimized Flash-Attention2 (Dao, 2023). In contrast, the training hour saving is relatively clear. For example, LongLoRA spends 56.6% training hours as that of LoRA in the 65536 context length.

In Table 13, we present the effects of $S^2$-Attn without Flash-Attention2 (Dao, 2023). LoRA$^+$ is included in this ablation. It shows that $S^2$-Attn achieves more speedup than that in Table 12. Without the help of Flash-Attention2 (Dao, 2023), the full attention baseline encounters *OOM* at the 16384 context fine-tuning in an $8\times$ A100 machine, while $S^2$-Attn is sufficient for this.

## B.6 SUPERVISED FINE-TUNING.

We further conducted supervised fine-tuning on ours to improve their QA ability. Although the models fine-tuned with Redpajama (Computer, 2023) present good perplexities, their chat ability is limited. We collect some question-answer pairs, relating to the materials like technical papers, science

---

[1] https://github.com/zhijian-liu/torchprofile

Table 12: Efficiency comparison on training hours and GPU memory cost. We fine-tune Llama2 (Touvron et al., 2023b) 7B model for 1000 iterations on $8\times$ A100 GPUs. We set batch size per GPU as 1 and gradient accumulation steps as 8. *OOM* means out of GPU memory. Flash-Attention2 (Dao, 2023) and DeepSpeed (Rasley et al., 2020) in stage 2 are included in these experiments. LongLoRA requires significantly lower computational overhead than fine-tuning the full model. It also demands fewer training hours compared to LoRA (Hu et al., 2022). Furthermore, the plain LoRA (Hu et al., 2022) fails to maintain the same level of accuracy as full fine-tuning when handling longer contexts.

| Training setting | 8192 | | 16384 | | 32768 | | 65536 | |
|---|---|---|---|---|---|---|---|---|
| | Train hours | Memory (GB) | Train hours | Memory (GB) | Train hours | Memory (GB) | Train hours | Memory (GB) |
| Full FT | 7.4 | 46.3 | 16.3 | 57.4 | 39.8 | 68.8 | *OOM* | |
| LoRA | 6.0 | 25.7 | 14.0 | 34.7 | 36.5 | 46.5 | 92.5 | 71.1 |
| LongLoRA | **5.2** | **25.6** | **11.3** | **34.6** | **24.6** | **46.4** | **52.4** | **69.8** |

Table 13: The efficiency effects of $S^2$-Attn without Flash-Attention2 (Dao, 2023). The fine-tuning settings are the same to Table 12. LoRA$^+$ is used. Without Flash-Attention2 (Dao, 2023), $S^2$-Attn improves the training speed by $2.1\times$ and GPU memory cost by $1.8\times$ on 8192 context length. Without $S^2$-Attn and Flash-Attention2, Llama2 7B can not be extended to 16384 context, due to *OOM*.

| $S^2$-Attn | 8192 | | 16384 | |
|---|---|---|---|---|
| | Train hours | Memory (GB) | Train hours | Memory (GB) |
| ✗ | 17.5 | 55.5 | *OOM* | |
| ✓ | **8.2** | **30.3** | **20.8** | **57.1** |

fiction, and other books. We have already filter out any potentially harmful or negative content in our training data. The questions we designed include summarization, relationships, and characters. We build the prompt format as the following line:

Below is {material_type}. Memorize the content and answer my question after the paper. {material_content} $n$ Now the material ends. {question}

{material_type} can be "book", "paper", and others. {material_content} is the long-context content in the document. {question} is the question we design. These questions can be some commonly used ones, like summarization and limitation. Or they can be specific to the material, like the question that is related to some roles in the book. We named our long-context instruction following dataset as LongAlpaca-12k, which contains 9k long-context QAs and 3k short QAs sampled from the original Alpaca data.

For SFT, we use the same learning rate, weight decay, and batch sizes as the context extension step. We train the models for 5 epochs. In the following, we provide some example questions and the answers from our model, in Figure 8 and Figure 9. Note that these example questions are not in the training set.

Table 14: Perplexity evaluation on PG19 (Rae et al., 2020) test split. We fine-tune Llama2 (Touvron et al., 2023b) in 7B and 13B sizes with 8192, 16384, and 32768 context lengths.

| Size | Training Context Length | LongLoRA S²-Attn | LoRA⁺ | Evaluation Context Length 2048 | 4096 | 8192 | 16384 | 32768 |
|---|---|---|---|---|---|---|---|---|
| 7B | 8192 | | | 7.55 | 7.21 | 6.98 | - | - |
| | | ✓ | | 7.53 | 7.20 | 7.01 | - | - |
| | | ✓ | ✓ | 7.70 | 7.35 | 7.14 | - | - |
| | 16384 | ✓ | | 7.56 | 7.21 | 6.97 | 6.80 | - |
| | | ✓ | ✓ | 7.65 | 7.28 | 7.02 | 6.86 | - |
| | 32768 | ✓ | | 7.76 | 7.36 | 7.09 | 7.04 | 7.03 |
| | | ✓ | ✓ | 8.29 | 7.83 | 7.54 | 7.35 | 7.22 |
| 13B | 8192 | ✓ | | 6.95 | 6.60 | 6.43 | - | - |
| | | ✓ | | 6.94 | 6.63 | 6.45 | - | - |
| | | ✓ | ✓ | 7.03 | 6.73 | 6.58 | - | - |
| | 16384 | ✓ | | 6.90 | 6.58 | 6.37 | 6.22 | - |
| | | ✓ | ✓ | 7.05 | 6.70 | 6.47 | 6.31 | - |
| | 32768 | ✓ | | 7.14 | 6.76 | 6.52 | 6.39 | 6.36 |
| | | ✓ | ✓ | 7.14 | 6.78 | 6.55 | 6.38 | 6.29 |

Table 15: Perplexity evaluation on PG19 (Rae et al., 2020) test split with the maximum context length that we can fine-tune on a single $8\times$ A100 machine. The Llama2 (Touvron et al., 2023b) models are fine-tuned on RedPajama (Computer, 2023).

| Size | Training Context Length | Evaluation Context Length 2048 | 4096 | 8192 | 16384 | 32768 | 65536 | 100,000 |
|---|---|---|---|---|---|---|---|---|
| 7B | 100,000 | 8.38 | 7.90 | 7.57 | 7.33 | 7.16 | 7.06 | 7.04 |
| 13B | 65536 | 7.63 | 7.21 | 6.94 | 6.75 | 6.62 | 6.57 | - |
| 70B | 32768 | 5.93 | 5.63 | 5.44 | 5.32 | 5.27 | - | - |

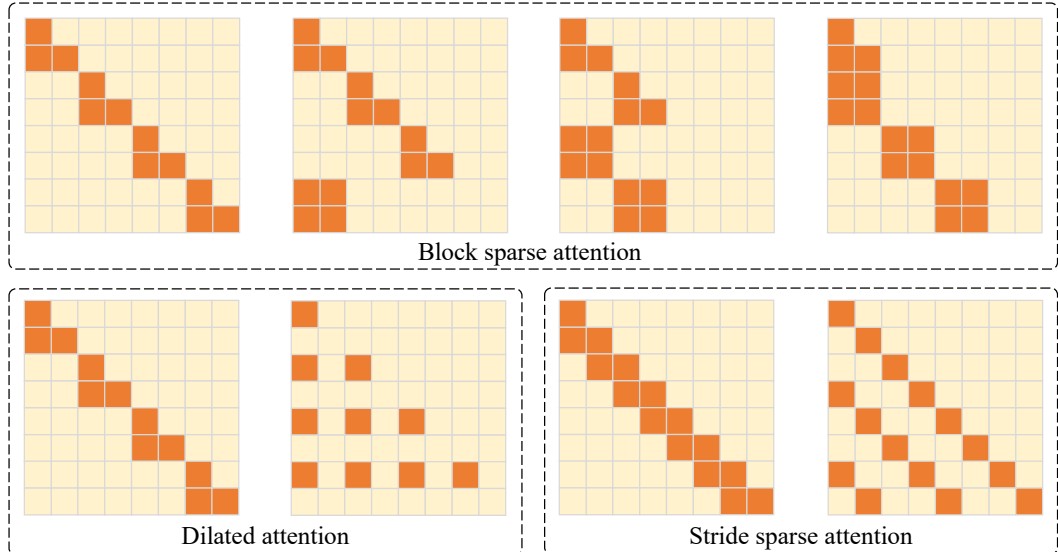

Figure 7: Illustration on alternative sparse attention patterns discussed in the paper. We modify the original block-wise sparse attention (Qiu et al., 2020) to be causal. For block sparse attention, we move its right-up blocks to left down, because of the causal mask. For stride sparse attention, we switch the patterns between local attention and stride attention. The local size is $\frac{n}{4}$ and the stride interval is $\sqrt{n}$, following (Child et al., 2019). We make sure that all alternative designs have similar amount of computation cost compared with S²-Attn.

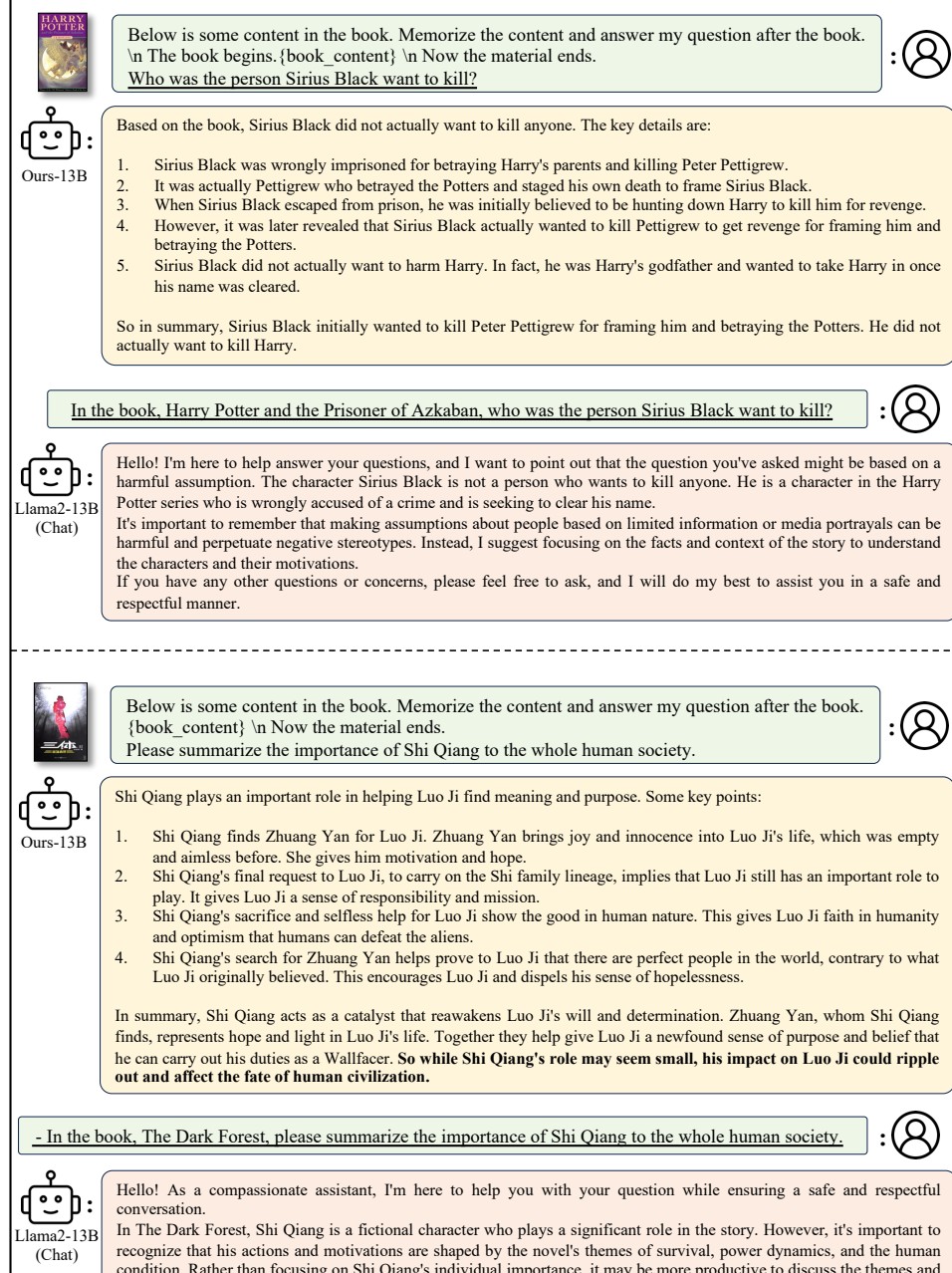

Figure 8: Examples on book-related questions [1][2]. We compare the answers from our 13B model with SFT and the chat version of Llama2 13B. During the pre-training stage of Llama2, some books might be used as training data. To ablate this, we ask the Llama2 13B chat model with the book title and the question. We provide our model with the related book sections. Our model produces more structured and concise answers, referring to detailed context, than those from Llama2 without long documents.

---

[1] https://en.wikipedia.org/wiki/Harry_Potter_and_the_Prisoner_of_Azkaban
[2] https://en.wikipedia.org/wiki/The_Dark_Forest

---------------------------------------- Contribution ----------------------------------------

Below is a paper. Memorize the material and answer my question after the paper.
{paper}\n Now the material ends.
What are the main contributions and novelties of this work?

The paper proposes an Action Units Relation Learning framework that consists of two main parts:
1. The Action Units Relation Transformer (ART) that models the relations between different facial action units (AU) at the AU-agnostic patches and the AU-specific regions. This helps capture intra-face relations that are useful for forgery detection.
2. The Tampered AU Prediction (TAP) that tampers the AU-related regions and provides Local Tampering Supervision. This helps improve the generalization ability of the model to unseen manipulation methods.

The key contributions are:
1. Modeling the relations between different AUs at the AU-agnostic patches using the ART encoder. This helps capture intra-face relations that are useful for forgery detection.
2. Generating challenging pseudo samples for the model to learn by tampering the AU-related regions using the TAP process. This helps improve the generalization ability of the model to unseen manipulation methods.
3. Achieving state-of-the-art performance on cross-dataset and cross-manipulation evaluations, demonstrating the effectiveness and generalization of the proposed framework.
4. Providing qualitative visualizations of the tampered regions using the predicted patch tokens, which helps interpret how different regions are modified in the manipulation process.

In summary, the main contributions are the proposed ART encoder that models intra-face relations and the TAP process that generates challenging pseudo samples, which together improve the generalization of the deepfake detection model.

---------------------------------------- Limitation ----------------------------------------

Below is a paper. Memorize the material and answer my question after the paper.
{paper}\n Now the material ends.
What are some limitations of the proposed 3DGNN method?

Possible answers:

1. The 3DGNN method relies on the initial graph construction based on 3D points and nearest neighbors. The accuracy of this graph construction can affect the performance of the model. The graph construction method used may not be optimal for all scenes.
2. The 3DGNN model uses a fixed number of propagation steps. The optimal number of propagation steps may vary for different scenes and objects. Using a fixed number of steps may not be the best approach.
3. The 3DGNN model struggles with objects that have similar depths or complex shapes, as discussed in the failure case analysis. The model's performance relies on the 3D geometric context, which may not be sufficient for some types of objects.
4. The model requires depth information from an RGBD sensor, which limits its applicability to scenes where depth data is not available. The model cannot be directly applied to 2D RGB images.
5. The model performance may degrade for very small objects, as shown in the analysis of performance gain by object size. The model seems better suited for medium and large objects.

---------------------------------------- Summarization ----------------------------------------

Below is a paper. Memorize the material and answer my question after the paper.
{paper}\n Now the material ends.
Please summarize the paper in one paragraph.

The paper proposes a new dataset called SAFECONV for the research of conversational safety. The dataset annotates unsafe spans in utterances and provides safe alternative responses to replace unsafe responses. The annotated unsafe spans help explain why an utterance is unsafe and provide guidance for generating safe responses. The dataset contains unsafe spans, unsafe responses, and safe alternative responses for over 100,000 dialogues from social media platforms. The paper compares SAFECONV with other datasets and shows that SAFECONV is more comprehensive. SAFECONV demonstrates that identifying unsafe spans can well explain the detection of unsafe utterances, and rewriting unsafe responses with context can mitigate a large proportion of unsafe behavior in chatbots. The dataset and models are released to advance the research of conversational safety.

Figure 9: Examples on paper (Ahn et al., 2023; Qi et al., 2017; Zhang et al., 2023) and questions related to contributions, limitations, and summarizations.

