# OpenReview forum: "LongLoRA: Efficient Fine-tuning of Long-Context Large Language Models"
_ICLR.cc/2024/Conference — ICLR 2024 oral_

### Official Review · Reviewer_WV76 · 2023-10-27

**Soundness:** 3 good
**Presentation:** 3 good
**Contribution:** 2 fair
**Rating:** 6
**Confidence:** 3

**Summary:**

this paper proposes some computationally efficient methods to continue finetuning a pretrained model to support longer context. The paper proposed a modification for localized attention to support longer context by shifting the subgroups during finetuning. The paper also experimented with LoRA for long-context adaptation.

**Strengths:**

1. the paper is well written and easy to follow. the proposed approach is a simple method that can adapt LLM for longer context without too much compute.
2. the paper has good ablation to show that LoRA on embedding and normalization is important for long-context adaptation.

**Weaknesses:**

1. the paper only evaluated on retrieval and perplexity. It would be good to evaluate on other generative tasks that require longer context.
2. the improvement on perplexity doesn't seem super consistent in Table. 4

**Questions:**

1. Have you tried evaluating on any generative tasks?

---

> ### Author Response · Authors · 2023-11-18
> **Response to Reviewer WV76**
>
> We are truly appreciated for your valuable comments. In the following, we provide responses to the concerns.
>
> **Q1: “Other generative tasks that require longer context.”**
>
> A: Thanks for your suggestion! We have included the evaluation on LongBench[1] and L-Eval [2], as shown in Table 9 and Table 10 in the revision. We fine-tuned Llama2 7B model with our long QA data. In these benchmarks, there are comprehensive generative tasks, including document QA, summarization, few-shot learning, code completion synthetic tasks, and other open-ended tasks in L-Eval. Our model presents comparable or even better performance than other counterparts, with about 4 hours for fine-tuning on 8 A100 GPUs. It takes 60 million tokens per epoch, 5 epochs, and 0.3 billion tokens in total for the supervised fine-tuning.
>
> Table 1 - Evaluation on LongBench English tasks.
>
> | Model | Avg | Single-Doc QA | Multi-Doc QA | Summarization | Few-shot Learning | Code | Synthetic |
> | --- | --- | --- | --- | --- | --- | --- | --- |
> | GPT-3.5-Turbo | 44.0 | 39.8 | 38.7 | 26.5 | 67.1 | 54.1 | 37.8 |
> | Llama2-7B-chat | 31.0 | 24.9 | 22.6 | 24.7 | 60.0 | 48.1 | 5.9 |
> | LongChat-v1.5-7B | 34.3 | 28.7 | 20.6 | 26.7 | 60.0 | 54.1 | 15.8 |
> | Vicuna-v1.5-7B | 31.9 | 28.0 | 18.6 | 26.0 | 66.2 | 47.3 | 5.5 |
> | Ours-7B | 36.8 | 28.7 | 28.1 | 27.8 | 63.7 | 56.0 | 16.7 |
>
> Table 2 - Evaluation on L-Eval open-ended tasks, i.e., comparing models to GPT-3.5-Turbo and judging win rates via GPT-4.
>
> | Model | Win-rate | Wins | Ties |
> | --- | --- | --- | --- |
> | LongChat-7B | 33.68 | 36 | 56 |
> | LongChat-v1.5-7B | 33.59 | 38 | 53 |
> | Vicuna-v1.5-7B | 25.52 | 22 | 54 |
> | Ours-7B | 39.06 | 45 | 60 |
>
> **Q2: “The improvement on perplexity in Table 4.”**
>
> A: Sorry for the confusion caused. The original Table 4 (Table 3 in the revision) is not designed to show perplexity improvements, which we never claim. The goal of LongLoRA is to significantly improve the training efficiency (in terms of both memory consumption and training speed) without compromising the perplexity achieved through finetuning the full model. We have clarified this point in the caption of Table 3 In the revision.
>
> [1] Yushi Bai, Xin Lv, Jiajie Zhang, Hongchang Lyu, Jiankai Tang, Zhidian Huang, Zhengxiao Du, Xiao Liu, Aohan Zeng, Lei Hou, Yuxiao Dong, Jie Tang, Juanzi Li: LongBench: A Bilingual, Multitask Benchmark for Long Context Understanding. CoRR abs/2308.14508 (2023)
>
> [2] Chenxin An, Shansan Gong, Ming Zhong, Mukai Li, Jun Zhang, Lingpeng Kong,  Xipeng Qiu: L-Eval: Instituting Standardized Evaluation for Long Context Language Models. CoRR abs/2307.11088 (2023)

---

### Official Review · Reviewer_Sw6W · 2023-11-01

**Soundness:** 4 excellent
**Presentation:** 4 excellent
**Contribution:** 4 excellent
**Rating:** 8
**Confidence:** 4

**Summary:**

This paper presents a method to perform LLM context extension with less memory and wall-clock time than existing methods. Their main modifications to improve efficiency are (1) training on local rather than global attention using the shift-short attention pattern, (2) using LoRA, and (3) modifying the norm and embedding layers in addition to the self-attention and feed-forward layers. The resulting method performs similarly to full fine-tuning.

**Strengths:**

(1) The method seems useful and impactful, and the evaluation is thorough with strong results.

(2) The authors perform very thorough ablations and isolate key design decisions (attention shift, modifying the norm & embedding layers) that enable the method to match full fine-tuning.

(3) The paper is well-written.

**Weaknesses:**

No major weaknesses.

**Questions:**

(1) While this is somewhat outside the scope of this paper, I would be curious about comparisons to methods that involve training a long-context LM from scratch.

(2) I am a bit confused why regular LoRA and LoRA+ (Table 11) use the same amount of memory. Does S^2-Attn reduce memory usage as well, or only flops?

---

> ### Author Response · Authors · 2023-11-18
> **Response to Reviewer Sw6W**
>
> We are truly appreciated for your valuable comments. In the following, we provide responses to the concerns.
>
> **Q1: “Comparisons to methods that train long-context from scratch.”**
>
> A: Thanks for this question. We agree with you that training long-context from scratch could be an interesting direction. Nevertheless, training long-context LLMs from scratch is too computationally expensive and unaffordable for most researchers. LongLoRA provides an efficiency advantage of orders of magnitude compared to training long-context LLMs from scratch. For instance, Llama-2 models require training with 2 trillion tokens across hundreds of GPUs, whereas LongLoRA models are finetuned on about 2 billion tokens for 32k context length using just 8 A100 GPUs.
>
> Recently, Meta published a concurrent research paper, LLama2-Long [1]. In this paper, the authors empirically verified that long-context continual training on short-context models is more efficient and similarly effective compared to pretraining from scratch with long sequences. This conclusion could further magnify the importance of LongLoRA.
>
> **Q2: “Regular LoRA and LongLoRA use the same amount of memory in Table 11. $S^2$-Attn for memory, or only ﬂops?”**
>
> A: The reason why $S^2$-Attn does not reduce the memory usage on top of LoRA (LoRA+) in the original Table 11 is that we adopt FlashAttention-2 [2] for the implementation of self-attention layers. This library fuses all operators in multi-head self-attention (MHSA) into a single CUDA kernel and thereby avoids writing the attention score matrix to DRAM. So the memory space required for MHSA is around $2 * B * N * C$, where B is the batch size, N is the sequence length and C is the number of channels. The multiplier 2 corresponds to input and output activations.
>
> Nevertheless, if FlashAttention-2 is not used, the vanilla dense attention requires $2 * B * N * C + B * N^2 * C$ memory space. Here, $B * N^2 * C$ corresponds to attention scores. In comparison, our $S^2$-Attn requires only $2 * B * N * C + 4 * B * (N/4)^2 * C = 2 * B * N * C + 1/4 * B * N^2 * C$ memory space. Empirically, we include an additional comparison that disables the FlashAttention-2 in the table below. Under a context length of 8192, $S^2$-Attn improves the training speed by 2.1x and memory usage by 1.8x. With a 16k context length, the original dense attention runs out of memory during training while $S^2$-Attn is still feasible. We have included this comparison in the Table 13 of the revision.
>
> | $S^2$-Attn | Train hours (8192) | Memory - GB (8192) | Train hours (16384) | Memory - GB (16384) |
> | --- | --- | --- | --- | --- |
> | x | 17.5 | 55.5 | OOM | OOM |
> | √ | 8.2 | 30.3 | 20.8 | 57.1 |
>
> [1] Wenhan Xiong, Jingyu Liu, Igor Molybog, Hejia Zhang, Prajjwal Bhargava, Rui Hou, Louis Martin, Rashi Rungta, Karthik Abinav Sankararaman, Barlas Oguz, Madian Khabsa, Han Fang, Yashar Mehdad, Sharan Narang, Kshitiz Malik, Angela Fan, Shruti Bhosale, Sergey Edunov, Mike Lewis, Sinong Wang, Hao Ma: Effective Long-Context Scaling of Foundation Models. CoRR abs/2309.16039 (2023)
>
> [2] Tri Dao: FlashAttention-2: Faster Attention with Better Parallelism and Work Partitioning. CoRR abs/2307.08691 (2023)

---

> > ### Comment · Reviewer_Sw6W · 2023-11-23
> >
> > Thanks for the great answers to my questions.

---

### Official Review · Reviewer_FsJg · 2023-11-01

**Soundness:** 3 good
**Presentation:** 2 fair
**Contribution:** 2 fair
**Rating:** 8
**Confidence:** 3

**Summary:**

This paper introduces a novel approach to extend the context length of transformer-based language models. The approach consists of two main ideas: 1) split the context into smaller subgroups and conduct attention in each group individually; 2) adapt the model to make use of this new attention approach via parameter-efficient fine-tuning with LoRA.

The authors conduct experiments with the Llama2 model family using models with 7B, 13B, and 70B parameters and compare their newly proposed approach to several baselines. In terms of perplexity, their proposed approach is able to maintain performance even when extending the context size by a factor of 16.

Beyond language modelling, the authors evaluate their method in a retrieval setup (finding a hidden key in a long sequence of text), demonstrating its improved performance over baselines.

**Strengths:**

- The proposed method builds on previous work and shows strong empirical results on long lange language modelling and a retrieval task
- The proposed approach is conceptually simple and can be implemented in a few lines of code (as demonstrated by the authors)
- The proposed approach can be combined with existing approaches for context extension such as positional interpolation
- The authors provide a detailed discussion of related work

**Weaknesses:**

- The efficiency aspect of the could could be more prominently discussed in the main body of the paper
- The presentation of the work could be improved. See below for suggestions

**Questions:**

**Presentation**

- I had difficulties understanding Figure 3. It would help if you add indices and annotations to the matrices in this plot. Additionally, it could be helpful to draw a (visual) connection between the blue matrices on the left and the attention patterns on the right.
- Be more consistent about the usage of $S^2$, shift short attention, LoRA+, and LongLoRA. Make it more explicit that LongLoRA = $S^2$ attention + LoRA.
- Table 7 is a great candidate for a line plot.
- When pointing to results in the Appendix, make sure to reference a specific section in the Appendix.
- The "attention patterns" ablation feels repetitive. How is it different from the "consistency to full attention" discussion in Section 3.2?
- In the section on retrieval-based evaluation you mention that your model is "somehow" able to handle longer context. What does this mean?

**Experiments**

- You mention several times that the original standard self-attention can be retained at inference time. It would be helpful to provide more details on that. Also, Table 2 is mentioned as evidence for that. It would be helpful to elaborate more about the results in this table.
- Table 3: What about an additional baseline that trains LoRA + embeddings?

---

> ### Author Response · Authors · 2023-11-18
> **Response to Reviewer FsJg**
>
> We are truly appreciated for your valuable comments. In the following, we provide responses to the concerns.
>
> **Q1: About Figure 3**
>
> A: Thanks for your constructive suggestion. We have updated Figure 3 in the revision, with indices and annotations to the matrices, and the visual connections between matrices and attention patterns.
>
> **Q2: “Be more consistent about $S^2$-Attn, shift short attention, LoRA+, and LongLoRA.”**
>
> A: We have updated these terms to be more consistent in the revision. We make more explicit that LongLoRA is the combination of $S^2$-Attn and LoRA+ in the abstract.
>
> **Q3: “Table 7 is a great candidate for a line plot.”**
>
> A: We have transformed the original Table 7 into a line plot. Please refer to Figure 5 in the revision.
>
> **Q4: “Reference a speciﬁc section in the Appendix”**
>
> A: In the revision, we have added specific section numbers in the appendix when we reference them in the paper.
>
> **Q5: “The attention patterns ablation feels repetitive to Section 3.2”**
>
> A: Thanks for the reminder! To avoid repetition, we have relocated the original Table 2 to Table 6 (in Section 4.2) and shortened relevant descriptions in Section 3.2. In Section 3.2, the objective is to show that we can “train sparse, test dense” with $S^2$-Attn. In Section 4.2, we further demonstrate that $S^2$-Attn achieves superior performance to existing sparse attention patterns.
>
> **Q6: “Retrieval-based evaluation - able to handle longer context.”**
>
> A: Sorry for the confusion caused. We have removed the word “somehow” in the section you mentioned. Retrieval-based evaluation shows that LongLoRA extends the context window of a pre-trained LLM. Specifically, Llama-2-7b sharply fails to retrieve the passkey when the context window exceeds 4k, while our method maintains a high retrieval accuracy (60-90%) even at a much longer context length of 33k-45k.
>
> **Q7: “The original standard self-attention at inference time.”**
>
> A: In Table 6 of the revision (the original Table 2), for each attention pattern, we evaluate its performance under two protocols. In the first row, we use sparse attention in both training and testing. In the second row, we use sparse attention only for training and the standard full attention for testing. For our $S^2$-Attn, it achieves the best perplexity with the original full self-attention at inference time. We have elaborated on these details in the table caption in the revision.
>
> **Q8: “An additional baseline that trains LoRA + embeddings.”**
>
> A: Thanks for your reminder. We have conducted this ablation and included it in Table 2 in the revision. Finetuning the embedding layer brings larger benefits than normalization layers.
>
> | Embed | x | √ | √ |
> | --- | --- | --- | --- |
> | Norm | √ | x | √ |
> | PPL | 10.49 | 8.29 | 8.12 |

---

> > ### Comment · Reviewer_FsJg · 2023-11-22
> > **Thanks for the update; raised score**
> >
> > I thank the authors for their detailed answers to my questions/suggestions.
> >
> > In the light of the author response to my questions, the other reviews, and the authors' response to these, I decided to increase my score from 6 to 8.

---

> > > ### Author Response · Authors · 2023-11-22
> > > **Thanks for your reply!**
> > >
> > > We are really grateful for your constructive comments and the reply. The comments indeed help us improve the paper greatly.

---

### Official Review · Reviewer_7K1u · 2023-11-11

**Soundness:** 3 good
**Presentation:** 2 fair
**Contribution:** 3 good
**Rating:** 6
**Confidence:** 4

**Summary:**

The authors propose a new method for adapting pretrained large language models (LLMs) to longer sequence lengths with a focus on efficiency. Prior works are either costly, due to requiring full fine-tuning of the language model or loose performance. The authors show that combining Low Rank Adaptation (LoRA) with sparse local attention provides improves efficiency while preserving performance. For LoRA, the authors note that a simple and cheap modification to LoRA, un-freezing embedding and normalization layer parameters, can prevent LoRA from loosing performance as sequence lengths increases. For sparse local attention, they employ a simple heuristic of splitting attention into independent groups of 2048 tokens. By overlapping groups within each layer at different attention heads, they ensure information flow between groups and are able to preserve performance at a level close to the much costlier full attention.

**Strengths:**

- The authors propose an extremely simple method, that performs well and is applicable to existing pretrained models

**Weaknesses:**

- The authors only evaluate perplexity and retrieval setting

**Questions:**

- Have you done experiments / ablation on optimal group size for different target sequence lengths? It seems you have derived that setting the group size to 25% of target sequence length is reasonable for 8192 sequence length, but it is unclear whether this 25% heuristic or a constant group size translates to longer sequence lengths.
- There are multiple ways to estimate model flops. Please provide the method / formula you used for Table 10.

---

> ### Author Response · Authors · 2023-11-18
> **Response to Reviewer 7K1u**
>
> We are truly appreciated for your valuable comments. In the following, we provide responses to the concerns.
>
> **Q1: “Evaluation only on perplexity and retrieval setting.”**
>
> A: We include additional comparisons on LongBench [1] and L-Eval [2] benchmarks. We fine-tune Llama2 7B with our method on our long QA data. We compare our model with GPT-3.5-Turbo and other Llama2-based long-context models, Vicuna and LongChat models, in the tables below. It shows that our 7B model presents comparable or even better performance than these Llama2-based long-context models, while ours only takes about 4 hours, about 0.3 billion tokens, for supervised fine-tuning on a single 8x A100 machine. We have included these evaluations in Table 9 and Table 10 in the revision.
>
> Table 1 - Evaluation on LongBench English tasks
>
> | Model | Avg | Single-Doc QA | Multi-Doc QA | Summarization | Few-shot Learning | Code | Synthetic |
> | --- | --- | --- | --- | --- | --- | --- | --- |
> | GPT-3.5-Turbo | 44.0 | 39.8 | 38.7 | 26.5 | 67.1 | 54.1 | 37.8 |
> | Llama2-7B-chat | 31.0 | 24.9 | 22.6 | 24.7 | 60.0 | 48.1 | 5.9 |
> | LongChat-v1.5-7B | 34.3 | 28.7 | 20.6 | 26.7 | 60.0 | 54.1 | 15.8 |
> | Vicuna-v1.5-7B | 31.9 | 28.0 | 18.6 | 26.0 | 66.2 | 47.3 | 5.5 |
> | Ours-7B | 36.8 | 28.7 | 28.1 | 27.8 | 63.7 | 56.0 | 16.7 |
>
> Table 2 - Evaluation on L-Eval open-ended tasks, i.e., comparing models to GPT-3.5-Turbo and judging win rates via GPT-4
>
> | Model | Win-rate | Wins | Ties |
> | --- | --- | --- | --- |
> | LongChat-7B | 33.68 | 36 | 56 |
> | LongChat-v1.5-7B | 33.59 | 38 | 53 |
> | Vicuna-v1.5-7B | 25.52 | 22 | 54 |
> | Ours-7B | 39.06 | 45 | 60 |
>
> **Q2: “Ablation on group size for longer sequence lengths”**
>
> A: Thanks for your suggestion! We perform additional ablation experiments on group size using a context length of 16384, as depicted in the table below. The results indicate that setting the group size to 1/4 of the context length is optimal in terms of efficiency-accuracy tradeoff. We have included these results in Table 7 in the revision.
>
> | Context Length | Full | 1/2 | 1/4 | 1/6 | 1/8 |
> | --- | --- | --- | --- | --- | --- |
> | 8192 | 8.02 | 8.04 | 8.04 | 8.10 | 8.16 |
> | 16384 | 7.82 | 7.84 | 7.86 | 7.94 | 7.98 |
>
> **Q3: “The method to estimate model FLOPs.”**
>
> A: We profile the context stage FLOPs of Llama2-7B using a batch size of 1 and various context lengths using a third-party tool, torchprofile [3]. The tool traces the computation graph and sums up the FLOPs of each node in the graph (e.g. Q/K/V/O projections, multi-head self-attention, fully-connected layers, and normalization layers).
>
> [1] Yushi Bai, Xin Lv, Jiajie Zhang, Hongchang Lyu, Jiankai Tang, Zhidian Huang, Zhengxiao Du, Xiao Liu, Aohan Zeng, Lei Hou, Yuxiao Dong, Jie Tang, Juanzi Li: LongBench: A Bilingual, Multitask Benchmark for Long Context Understanding. CoRR abs/2308.14508 (2023)
>
> [2] Chenxin An, Shansan Gong, Ming Zhong, Mukai Li, Jun Zhang, Lingpeng Kong,  Xipeng Qiu: L-Eval: Instituting Standardized Evaluation for Long Context Language Models. CoRR abs/2307.11088 (2023)
>
> [3] torchprofile. https://github.com/zhijian-liu/torchprofile

---

### Author Response · Authors · 2023-11-18
**General Response**

We thank all reviewers for the detailed and constructive reviews. We have revised the paper based on the comments. Here are some highlights in the paper revision.

1. We provide additional evaluations on the LongBench and L-Eval benchmarks.
2. We have conducted additional ablations, including group size for longer context length, LoRA + embeddings baseline, and efficiency comparison without FlashAttention-2.
3. We have made additional modifications and clarifications and double-check the paper writing for a better paper presentation. We highlight the revised parts in blue.

In the following, we respond to all concerns one by one.

---

### Meta-Review · Area_Chair_ZZUi · 2023-12-06

**Metareview:**

The paper presents LongLoRA, an efficient fine-tuning approach for extending the context sizes of pre-trained large language models (LLMs) with limited computational cost. The authors propose a combination of sparse local attention and Low Rank Adaptation (LoRA) to achieve this. The reviewers appreciate the simplicity of the proposed method and its applicability to existing pre-trained models. The empirical results presented are strong, and the authors have conducted thorough ablations to isolate key design decisions. However, the reviewers suggest that the authors could improve the presentation of their work and extend their evaluation to other generative tasks that require longer context. Overall, the reviewers recommend acceptance of the paper due to its novel approach and potential impact on the field.

**Justification For Why Not Higher Score:**

N/A

**Justification For Why Not Lower Score:**

The paper presents a novel and efficient approach to extend the context sizes of pre-trained large language models (LLMs), which is a good contribution to the field. The authors have demonstrated strong empirical results and conducted thorough ablations to support their claims. Despite some minor weaknesses pointed out by the reviewers, such as the need for improved presentation and extended evaluation, the strengths of the paper, including its simplicity, outweigh these concerns.

---

### Decision · Program_Chairs · 2024-01-16

Accept (oral)